# Construction and Modification of Topological Tables for Digital Models of Linear Complexes

Aleksandr N. Rozhkov [1,2,*] and Vera V. Galishnikova [1,3]

1   Department of Civil Engineering, Peoples' Friendship University of Russia, 117198 Moscow, Russia
2   Department of Basic Education, Moscow State University of Civil Engineering (National Research University), 129337 Moscow, Russia
3   Department of Information Science and Applied Mathematics, Moscow State University of Civil Engineering (National Research University), 129337 Moscow, Russia
*   Correspondence: rozhkovalex@hotmail.com

**Abstract:** Building information systems use topological tables to implement the transition from two-dimensional line drawings of the geometry of buildings to digital three-dimensional models of linear complexes. The topological elements of the complex are named and the topological relations of the complex are described by arranging the element names in topological tables. The efficient construction and modification of topological tables for complete buildings is investigated. The topology of a linear complex with nodes, edges, faces, and cells is described with 12 tables. Three of the tables of a complex are independent of each other and form a basis for the construction of the other tables. A highly efficient construction algorithm with complexity O (number of cells) for typical buildings with an approximately constant number of edges per face and faces per cell of is presented. In practice, building designs and their digital models are frequently modified. A modification algorithm is presented, whose complexity equals that of the construction algorithm. Examples illustrate that the efficient algorithms permit the replacement of the conventional focus on the topology of building components by a focus on the topology of the entire building. A set of properties of the original, which are not explicitly described by the topological tables, for example, the orientation of surfaces and multiply connected domains, are analyzed in the paper. An overview of the research dealing with the topological attributes that are not contained in topological tables concludes the paper.

**Keywords:** linear complex; cell; polyhedral partition; topological modeling; topological tables; topology; neighborhood

## 1. Introduction

It is a widely held view that the goal of achieving interoperability of the software packages of various vendors by introducing the Industry Foundation Classes has not been reached. The aim of our research is to make a new attempt to advance interoperability by making the description of the topology and the geometry of buildings with digital models explicit and complete. Information is called explicit if it is described with variables. Information that must be derived from data with algorithms is called implicit. Our research pursues several novel concepts. In addition, some aspects of the construction of building information models, for which solutions have been found in the past, are reconsidered and modified to suit our new point of view.

Buildings have been planned and constructed for centuries using two-dimensional line drawings showing scaled projections of the shape of buildings on flat surfaces. A projection is defined by the location of the observer, the direction of view and the location of the projection surface.

The contents of a typical line drawing is restricted to a subset of the components of the building. An elevation shows the components that are visible for an external observer with

a horizontal direction of view. The content of a plan is determined by placing a horizontal cut at a selected level of the building and showing the components which are visible to an observer with a downward direction of view if the part of the building above the cut is removed. The content of a cross-section is determined by placing a vertical cut through the building and showing the components that are visible to an observer with a horizontal direction of view if the part of the building on one side of the cut is removed. Similar procedures are applied to describe components of buildings with elevations, plans and cross-sections. Figure 1 shows typical line drawings of a simple building without floor and roof.

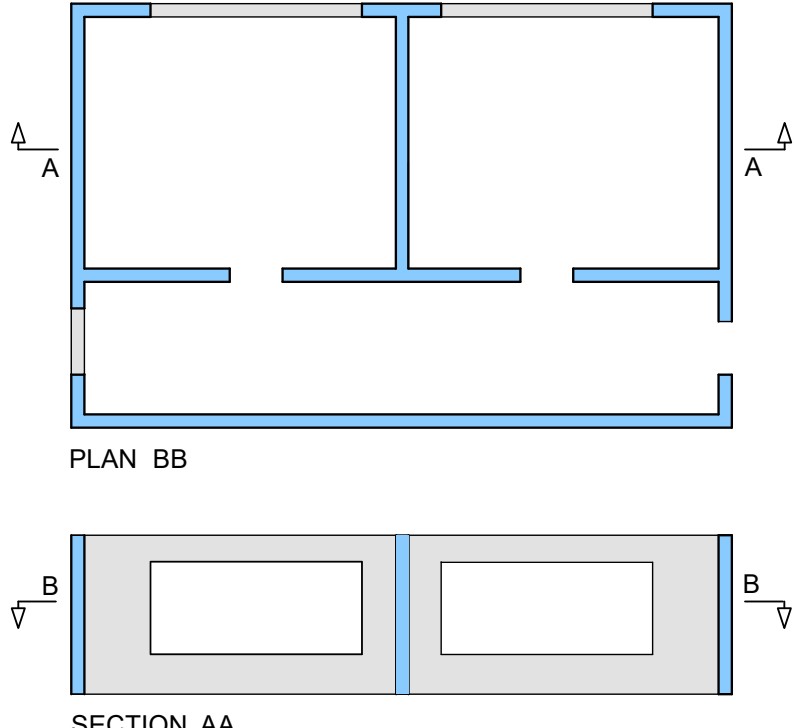

**Figure 1.** Line drawings showing a plan and a section of a building.

Information is transferred during the traditional planning and construction process by passing the set of drawings describing the building on paper from the author to the reader of the drawings. A trained reader uses his imagination to assemble a virtual three-dimensional model of the building from the two-dimensional information contained in the elevations, plans and cross-sections. The assembly process of the reader is not the inverse of the drawing process of the author.

The author selects some of the attributes of the three-dimensional original as explicit data in the two-dimensional line drawings. The included geometric data are incomplete and the topology is not explicit. For example, the drawings do not contain explicit data that relate a point in the plan to a specific point in an elevation or a section. Typically, a point in a plan corresponds to many points in an elevation or a section.

The reader adds implicit information to the explicit information of the line drawing to create his mental model of the original. Persons with different knowledge, skill and professional experience add different implicit information to the same set of line drawings, thus creating different mental models of the same original. The implicit information is a source of error, vagueness and inconsistency in engineering practice.

The aim of our research is to use the speed and the storage capacity of the digital environment to lessen the amount of implicit information in engineering. The two-dimensional line drawings describing the shape of buildings are replaced by digital three-dimensional models with exact topology and robust geometry. The mapping between original and model is bijective, such that the information and insight gained with the model can be applied to

the original and vice versa. Because the graphic computer interfaces are two-dimensional, and perspectives distort the shape of buildings, the three-dimensional computer model cannot be presented directly to the engineer. Therefore, conventional line drawings remain a valuable tool to visualize shape in engineering practice. However, the line drawings for the user are prepared upon demand from a complete three-dimensional computer model of the entire building.

In our digital approach, the building is described by a three-dimensional model of the shape and location of the surfaces of its components and of the interfaces between its components. The model is called a linear complex if all of its surfaces and interfaces are flat and bounded by straight lines. Figure 2 shows the linear complex for the building in Figure 1. The domains of a linear complex are nodes, edges, surfaces and cells that are path-connected. An edge is a straight-line segment connecting its end nodes. A face is a flat area bounded by the edges of the polygonal curves of its boundary. A cell is a volume bounded by the faces of the polyhedral surfaces of its boundary.

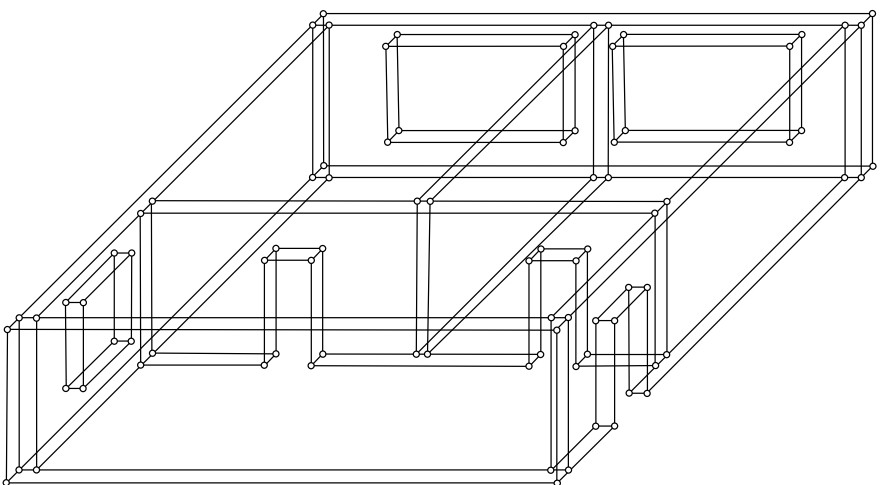

**Figure 2.** Linear complex for the building shown in Figure 1.

The geometry of a three-dimensional linear complex is specified with the coordinates of its nodes that are stored in a node table. The contact between the domains of the complex is described with the established concept of topological tables. The method is illustrated with the unit cube in Figure 3. Unique names are assigned to the domains of the complex: $n_1$ to $n_8$ for the nodes, $e_1$ to $e_{12}$ for the edges, $f_1$ to $f_6$ for the faces and $c_1$ for the cell.

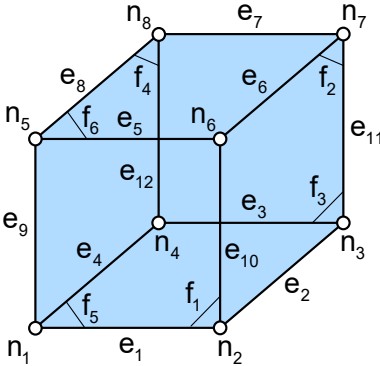

**Figure 3.** Topological domains of a unit cube $c_1$.

Several topological tables are constructed for a complex. For each table, a key domain type, for example, edge, and a value domain type, for example, node, are specified. The name of the table is the composite of the key and value types, for example, edge–node–table.

The first column of the table contains all instances of the key type, for example, edges $e_1$ to $e_{12}$ of the cube. The first column of each row of the table contains a key domain. The other columns contain value domains that are in contact with the key domain. For example, the row of the edge–node–table whose first column contains edge $e_3$ as key contains nodes $n_3$ and $n_4$ as values. Tables 1–4 show the 12 topological tables of the unit cube.

**Table 1.** Topological Tables (node–edge, node–face and node–cell) for a unit cube.

| Node–Edge–Table | | Node–Face–Table | | Node–Cell–Table | |
|---|---|---|---|---|---|
| **Node** | **Edges** | **Node** | **Faces** | **Node** | **Cells** |
| *n1* | *e1, e4, e9* | *n1* | *f1, f4, f5* | *n1* | *c1* |
| *n2* | *e1, e2, e10* | *n2* | *f1, f2, f5* | *n2* | *c1* |
| *n3* | *e2, e3, e11* | *n3* | *f2, f3, f5* | *n3* | *c1* |
| *n4* | *e3, e4, e12* | *n4* | *f3, f4, f5* | *n4* | *c1* |
| *n5* | *e5, e8, e9* | *n5* | *f1, f4, f6* | *n5* | *c1* |
| *n6* | *e5, e6, e10* | *n6* | *f1, f2, f6* | *n6* | *c1* |
| *n7* | *e6, e7, e11* | *n7* | *f2, f3, f6* | *n7* | *c1* |
| *n8* | *e7, e8, e12* | *n8* | *f3, f4, f6* | *n8* | *c1* |

**Table 2.** Topological Tables (edge–node, edge–face and edge–cell) for a unit cube.

| Edge–Node–Table | | Edge–Face–Table | | Edge–Cell–Table | |
|---|---|---|---|---|---|
| **Edge** | **Nodes** | **Edge** | **Faces** | **Edge** | **Cells** |
| *e1* | *n1, n2* | *e1* | *f1, f5* | *e1* | *c1* |
| *e2* | *n2, n3* | *e2* | *f2, f5* | *e2* | *c1* |
| *e3* | *n3, n4* | *e3* | *f3, f5* | *e3* | *c1* |
| *e4* | *n1, n4* | *e4* | *f4, f5* | *e4* | *c1* |
| *e5* | *n5, n6* | *e5* | *f1, f6* | *e5* | *c1* |
| *e6* | *n6, n7* | *e6* | *f2, f6* | *e6* | *c1* |
| *e6* | *n7, n8* | *e6* | *f3, f6* | *e6* | *c1* |
| *e8* | *n5, n8* | *e8* | *f4, f6* | *e8* | *c1* |
| *e9* | *n1, n5* | *e9* | *f1, f4* | *e9* | *c1* |
| *e10* | *n2, n6* | *e10* | *f1, f2* | *e10* | *c1* |
| *e11* | *n3, n7* | *e11* | *f2, f3* | *e11* | *c1* |
| *e12* | *n4, n8* | *e12* | *f3, f4* | *e12* | *c1* |

**Table 3.** Topological Tables (face–node, face–edge and face–cell) for a unit cube.

| Face–Node–Table | | Face–Edge–Table | | Face–Cell–Table | |
|---|---|---|---|---|---|
| **Face** | **Nodes** | **Face** | **Edges** | **Face** | **Cells** |
| *f1* | *n1, n2, n5, n6* | *f1* | *e1, e5, e9, e10* | *f1* | *c1* |
| *f2* | *n2, n3, n6, n7* | *f2* | *e2, e6, e10, e11* | *f2* | *c1* |
| *f3* | *n3, n4, n7, n8* | *f3* | *e3, e7, e11, e12* | *f3* | *c1* |
| *f4* | *n1, n4, n5, n8* | *f4* | *e4, e8, e9, e12* | *f4* | *c1* |
| *f5* | *n1, n2, n3, n4* | *f5* | *e1, e2, e3, e4* | *f5* | *c1* |
| *f6* | *n5, n6, n7, n8* | *f6* | *e5, e6, e7, e8* | *f6* | *c1* |

**Table 4.** Topological Tables (cell–node, cell–edge and cell–face) for a unit cube.

| Cell–Node–Table | | Cell–Edge–Table | | Cell–Face–Table | |
|---|---|---|---|---|---|
| **Cell** | **Nodes** | **Cell** | **Edges** | **Cell** | **Faces** |
| *c1* | *n1, n2, n3, n4, n5, n6, n7, n8* | *c1* | *e1, e2, e3, e4, e5, e6, e7, e8, e9, e10, e11, e12* | *c1* | *f1, f2, f3, f4, f5, f6* |

The matrix in Table 5 summarizes the topological tables of linear complexes.

**Table 5.** Matrix of the Topological Tables $T_{rs}$.

| Key Domains | Value Domains | | | |
|---|---|---|---|---|
| | s = 0: Nodes | s = 1: Edges | S = 2: Faces | s = 3: Cells |
| r = 0: node | X | $T_{01}$ | $T_{02}$ | $T_{03}$ |
| r = 1: edge | $T_{10}$ | X | $T_{12}$ | $T_{13}$ |
| r = 2: face | $T_{20}$ | $T_{21}$ | X | $T_{23}$ |
| r = 3: cell | $T_{30}$ | $T_{31}$ | $T_{32}$ | X |

## 2. Construction of the Dependent Topological Tables

The topological tables in Table 5 are not independent. Three independent tables form a base from which the other nine tables can be derived. The base tables can be chosen in several ways. In our research, the domain types are ranked according to their dimension from 0 for nodes to 3 for cells. The three base tables are chosen such that the rank of the key domain exceeds the rank of the value domains by 1. The rows of the key domains of the base tables therefore contain the following value domains:

- Edge–node–table: the row with key edge $e_j$ contains the nodes of edge $e_j$.
- Face–edge–table: the row with key face $f_k$ contains the edges of face $f_k$.
- Cell–face–table: the row with key cell $c_m$ contains the faces of cell $c_m$.

The three base tables define the contacts between the domains of the linear complex, whereas the other topological tables are constructed to support navigation in the complex. For example, the node–edge–table contains the edges of the complex that are in contact with a specified node, and the cell–node–table contains the nodes of a specified cell. An efficient algorithm has been developed for the construction of the dependent topological tables. The algorithm is based on the property that a base table with a key of rank n contains value domains of rank n − 1. The workflow in the algorithm consists of four nested loops, which operate on the basis tables to construct the dependent tables:

1. The outer loop 1 traverses the cells *c* of the complex in the first column of the cell–face–table.
2. The first inner loop 2 traverses the faces in row *c* of the cell–face–table and performs the following operations for each face *f*:

   - face *f* is added to row *c* of the cell–face–table (optional);
   - cell *c* is added to row *f* of the face–cell–table.

3. The second inner loop 3 traverses the edges in row *f* of the face-edge–table and performs the following operations for each edge *e*:

   - edge *e* is added to row *c* of the cell–edge–table;
   - edge *e* is added to row *f* of the face–edge–table (optional);
   - face *f* is added to row *e* of the edge–face–table;
   - cell *c* is added to row *e* of the edge–cell–table.

4. The third inner loop 4 traverses the nodes in row *e* of the edge–node–table and performs the following operations for each node *n*:

   - node *n* is added to row *c* of the cell–node–table;
   - node *n* is added to face *f* of the face–node–table;
   - node *n* is added to edge *e* of the edge–node–table (optional);
   - cell *c* is added to row *n* of the node–cell–table;
   - face *f* is added to row *n* of the node–face–table;
   - edge *e* is added to row *n* of the node–edge–table.

The operations, which are marked as optional, can be omitted in the algorithm because they construct the three base tables. The algorithm is conveniently implemented by storing each table in a map, using the key domain as key of the map entry and the set of value domains in the row of the table as value of the map entry.

It is noted that the algorithm attempts to put some domains into the same map more than once. For example, if a cell has two faces $f_1$ and $f_2$ with a common edge $e$, the algorithm attempts to enter edge $e$ in the cell–edge–map once when in traverses the edges of face $f_1$, and a second time when it traverses the edges of face $f_2$. The method, which adds domains to a table, must suppress multiple entry of the same domain. The algorithm has been implemented and tested on the Java platform.

The complexity of the algorithm that constructs the dependent topological tables is determined by counting the attempted number of domain additions to the dependent tables in the four loops. The optional operations not counted:

- Loop 1 is performed $N_c$ times, where $N_c$ is the number of cells in the complex.
- Loop 2 is performed $N_f$ times per cycle of loop 1, where $N_f$ is the average number of faces per cell. There is 1 addition per cycle of the loop.
- Loop 3 is performed $N_e$ times per cycle of loop 2, where $N_e$ is the average number of edges per face. There are 3 additions per cycle of the loop.
- Loop 4 is performed twice per cycle of loop 3, because each edge has two nodes. There are 5 additions per cycle of the loop.

The total number of attempted additions to the dependent topological tables is:

$$N_t = N_c \, N_f \, (1 + N_e \, (3 + 2 \times 5)) = N_c \, N_f \, (1 + 13N_e) \approx 13 \, N_c \, N_f \, N_e \qquad (1)$$

$N_t$—total number of attempted domain additions to the dependent tables.
$N_c$—number of cells in the linear complex.
$N_f$—average number of faces per cell.
$N_e$—average number of edges per face.

In typical buildings, the average number of faces per cell is approximately independent of the size of the building. Similarly, the average number of edges per face is nearly independent of the size of the building. The factors $N_f$ and $N_e$ can therefore be treated as constants in the complexity analysis, such that:

$$N_t \approx (13N_f \, N_e) \, N_c \approx const \times N_c \qquad (2)$$

The complexity of the algorithm that constructs the dependent topological tables from the three basis tables is $O(N_c)$ provided the assumption is satisfied that the number of faces per cell and the number of edges per face are constant.

## 3. Modification of Linear Complexes

The design of a building proceeds in many design cycles consisting of design steps. The design steps continuously modify the shape of the linear complex. The modifications of the complex require the following three types of modifications in its topological tables:

- New domains are added to the complex.
- Old values of attributes of domains are replaced by new values.
- Old domains are removed from the complex.

A design step consists of several modifications. For example, if a new face with new edges and new nodes is added in a design step, the new edges and nodes must be added in the same design step. Figure 4 shows a design step in which an octant is cut from the unit cube in Figure 3. The design step requires the removal of node $n_6$, changes of the attributes of edges $e_5$, $e_6$ and $e_{10}$ as well as faces $f_1$, $f_2$ and $f_6$, and addition of nodes $n_9$ to $n_{15}$, edges $e_{13}$ to $e_{21}$ as well as faces $f_7$ to $f_9$. The design step ends with the modification of the face set of cell $c_1$.

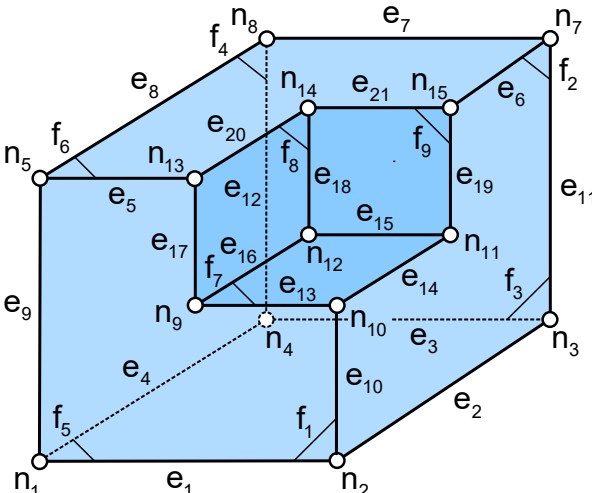

**Figure 4.** Unit cube after an octant has been removed in a design step.

Table 6 shows the face–node–table of the modified unit cube.

**Table 6.** Face–node–table for the modified unit cube.

| Face | Nodes |
|------|-------|
| f1 | n1, n2, n10, n9, n13, n5 |
| f2 | n2, n3, n7, n15, n11, n10 |
| f3 | n3, n7, n8, n4 |
| f4 | n4, n8, n5, n1 |
| f5 | n1, n2, n3, n4 |
| f6 | n5, n13, n14, n15, n7, n8 |
| f7 | n9, n10, n11, n12 |
| f8 | n9, n12, n14, n13 |
| f9 | n11, n14, n15, n12 |

At the start of a design step, the node table, the three basic topological tables and the domains objects in the database define the geometry and the contacts of the old domains of the complex. For every domain that is added to or removed from the complex, or whose attributes modified, the tables and the domain objects in the data base are treated as follows:

- If a new node, edge, face or cell is added to the complex, it is added to the node–table, the edge–node–table, the face–edge–table or the cell–face–table, respectively. Some of the attributes of the new domain can be old domains. The data base object of the new domain is added to the data base of the complex.
- If attributes of an old domain are changed, the old domain is retained in the node table or in the topological base table for the domain type. The new attributes replace the old attributes in the data base object of the domain.
- If an old node, edge, face or cell is removed from the complex, the domain is removed from the node–table, the edge–node–table, the face–edge–table or the cell–face–table, respectively. The data base object of the domain is removed from the data base of the complex.

At the end of the design step, the dependent topological tables are recomputed with the algorithm that is described above for the initial construction of the dependent tables. The complexity of the algorithm for the modification of the topological tables of a linear complex is therefore the same as the complexity of the algorithm for the construction if the initial tables. The algorithm has been implemented and tested on the Java platform.

Figure 5 shows a modification of the linear complex of the building in Figures 1 and 2. A part of room 3 is added to room 1 by removing the old wall between rooms 1 and 3 and extending the wall between rooms 1 and 2 such that it forms a new wall between rooms 1

and 3. The added domains are marked in red color in the figure. The removed domains and the modified domains are not marked.

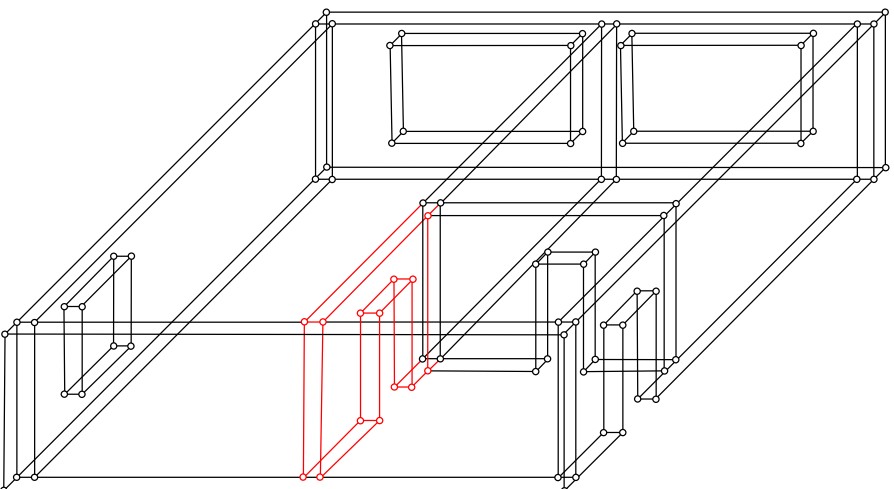

**Figure 5.** Modification of the linear complex in Figure 2.

## 4. Complete Topological Properties of Domains

The topological tables of a linear complex contain its contact data explicitly. The question arises as to whether the topological tables contain the complete topological information of the complex explicitly. Two examples are presented to show that this is not the case: the orientation of faces and cells and the multiple connectivity of faces and cells. The topological concepts, which are introduced in the examples, are described formally and in detail together with other novel concepts in references [1–3].

A row of the face–edge–table contains the edges of a face in arbitrary order. The explicit topological information for the boundary of the face is the polygonal curve in the left diagram of Figure 6. Each end node of each edge equals the start node of the edge with which it is in contact. The information for the polygon is implicit because it must be derived from the face–edge–table and the edge–node–table with an algorithm.

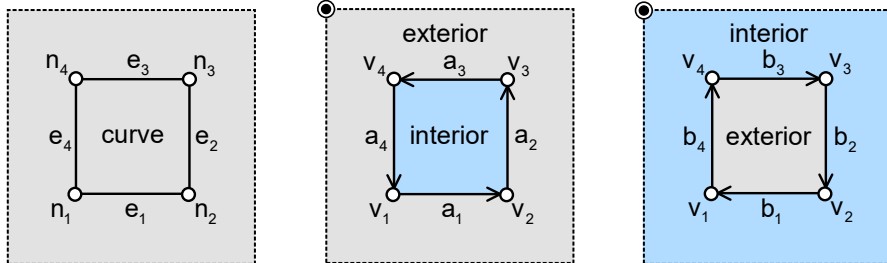

**Figure 6.** Faces with polygonal boundary curves.

The polygonal curve in the left diagram divides the plane containing the face into a bounded and an unbounded area. Conventionally, the polygonal curve is assumed to be the boundary of the bounded area. However, the polygonal curve is also the boundary of the unbounded area. The explicit data in the topological tables do not distinguish between these two cases.

The edges of a polygonal curve can be directed in the clockwise or in the anticlockwise direction around the normal vector of the plane containing the polygon. These two directions are used to distinguish the bounded and the unbounded area defined by the polygonal curve. A polygonal curve, whose direction is given, is called oriented. If a directed polygonal curve is traversed in the forward direction, the area of the polygon bounded by the curve lies to the left of the observer. A face with an oriented boundary

curve is called an oriented face. The center and right diagrams of Figure 6 show the two oriented faces and the directions of their boundary curves.

A row of the cell–face–table contains the faces of a cell in arbitrary order. The face set does not explicitly define the interior and the exterior of the polyhedron. The faces form a polyhedral surface that divides the space containing the surface into a bounded and an unbounded volume. Vectors that are normal to the surface have one of two possible directions, which are used to distinguish the two volumes defined by the surface. The normal vectors of the surface of an oriented polyhedron by definition point into the exterior of the polyhedron as shown in Figure 7.

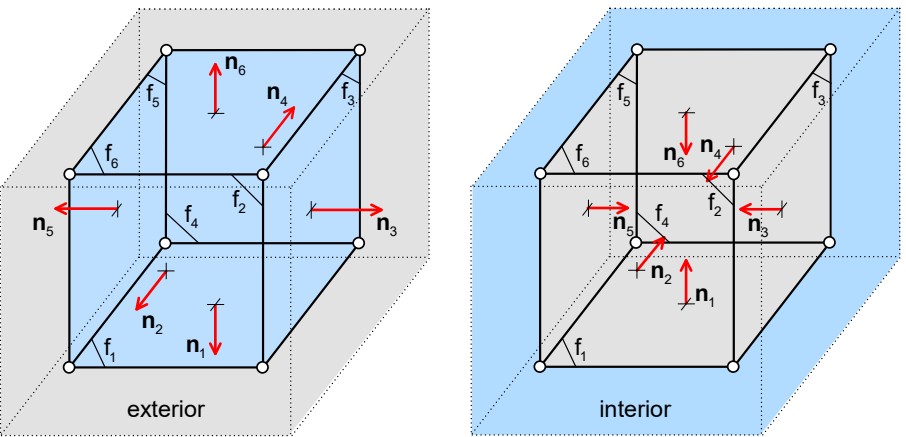

**Figure 7.** Oriented cells with polyhedral boundary surfaces.

The second example for topological attributes of linear complexes, which are not explicitly contained in topological tables, are multiply connected domains. A path-connected face is called multiply connected if it contains at least one hole. As a result, there exists a closed curve consisting of points of the face, which cannot be contracted continuously inside the face to a point. Multiply connected faces are not explicitly contained in topological tables.

A multiply connected face can artificially be reduced to a set of simply connected faces by triangulation, or by the famous method of cuts applied by Gauss. In our novel approach, multiply connected faces are treated as intersections of simply connected faces as shown in Figure 8. This concept becomes possible because the bounded and unbounded areas defined by a polyhedral curve can be distinguished by their orientation.

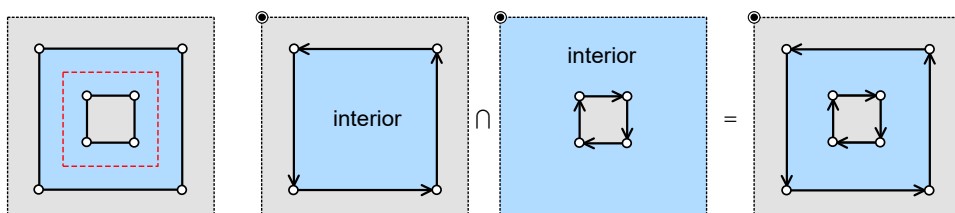

**Figure 8.** Multiply connected face.

A row of a cell–face–table contains the faces of a cell in arbitrary order. A cell is multiply connected if it contains at least one hole, such that its boundary consists of at least two closed polyhedral surfaces. The cell–face–table does not explicitly show whether the faces of a cell form a single surface or more than one surface. The topological tables therefore do not show explicitly whether a cell is simply connected or multiply connected (see Figure 9).

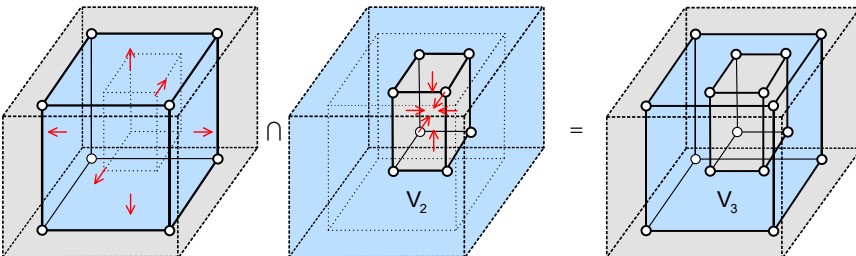

**Figure 9.** Multiply connected cell.

The examples show that the description of the topology of linear complexes with topological tables is incomplete. The classical theory of unions and intersections of convex polyhedra, whose roots go back over centuries, does not solve this problem. A significant amount of research has been conducted in the past decades to develop more complete and practicable concepts for the description of the topology of linear complexes. Major driving forces for the research are the demands in city planning, architectural and engineering practice for building information models [1–3] and geospatial data management [4,5]. Important progress in the development of topological concepts is documented in references [1–44] attached to this paper. The option of using relational data bases rather than the object-oriented methods presented by the authors is presented in references [4,5]. The main aspects of the progress are summarized and evaluated in Section 2 of reference [1]. A new topological paradigm based on partitions of spaces and the inclusion of imaginary domains is presented in this reference.

## 5. Conclusions

The investigation of topological tables for linear complexes has shown that these tables do not contain the complete explicit topological attributes of a linear complex. Two of the missing attributes, orientation and multiplicity, are presented in Section 4 of the paper. The question arises as to whether there are other topological attributes, which should be contained in the digital model as explicit data. A literature survey has been conducted to search for such attributes. Additional features such as construction by partitioning instead of assembly to avoid collisions and voids, robustness achieved by topologically controlled work steps, and models with unbounded as well as imaginary domains have been discovered or observed. Oriented polygons of edges in planes and dihedral cycles of faces at edges have been identified as the primary structural elements of linear complexes. The research is leading to a theory for partition models of linear complexes. Partition models contain the topological attributes explicitly.

**Author Contributions:** Conceptualization, methodology, software, A.N.R. and V.V.G.; validation, formal analysis, investigation, A.N.R. and V.V.G.; resources, data curation, A.N.R. and V.V.G.; writing—original draft preparation, A.N.R. and V.V.G.; writing—review and editing, A.N.R. and V.V.G.; visualization, A.N.R.; supervision, project administration, funding acquisition, V.V.G. All authors have read and agreed to the published version of the manuscript.

**Funding:** The research is funded by the Russian Foundation for Basic Research (RFBR)—Project Number 20-57-12006.

**Institutional Review Board Statement:** Not applicable.

**Informed Consent Statement:** Not applicable.

**Data Availability Statement:** The data presented in the research can be obtained by contacting the corresponding author.

**Conflicts of Interest:** The authors declare no conflict of interest.

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
