# Peer review of "Construction and Modification of Topological Tables for Digital Models of Linear Complexes"

_mca, doi:10.3390/mca28020037_

Round 1

Reviewer 1 Report

This is a very good paper, addressing its content in a very fundamental and theoretical manner, and yet presenting its very practical applications. 

The one mistake I found was in Table 5, where the second column heading should read as s=0; nodes (NOT "faces").

Perhaps the authors may care to engage (ONLY if they wish) with formal approaches in topology when describing Section 4 - e.g. convexity and concavity when tackling bounded and unbounded area; and the notion of holes when tackling multiply connected domains. I am not an expert on topolgy myself, but perhaps Section 4 would benefit by some references to work done in topology to tackle the above two examples described therein.

Author Response

The authors thank the reviewer for the evaluation of the submitted paper and for helpful comments and suggestions.

Following the written recommendations of the reviewers, as made available by the editor, the authors have modified the submitted paper and have made additions that have been suggested by the reviewers. In the attached of the modified paper, the modifications are marked in yellow color. The following remarks describe the considerations of the authors concerning the modifications and the additions.

1. The mistake in Table 5 has been corrected.

2. The reviewer suggests to add “formal approaches in topology when describing Section 4 - e.g. convexity and concavity when tackling bounded and unbounded area; and the notion of holes when tackling multiply connected domains”. 

The authors have adopted this suggestion and have added references [41], [42] and [43] to the list of publications. These references are introduced in Section 4. The added publications present the formal topological structure of linear partition models and their implementation. Novel approaches to convexity and concavity, to bounded and unbounded domains and to multiply connected faces and cells are treated in depth in these papers.

The authors were familiar through personal contacts with the research work that is described in the publications that have now been added to the paper. However, the authors were not able to refer to the research work of their colleagues in the submitted paper, because the articles of the colleagues had not yet been published at the time the authors submitted their paper. The references have now been added to the paper.

Reviewer 2 Report

Referee report for "Construction and Modification of Topological Tables for Digital Models of Linear Complexes"

The authors describe a way to store topological information of buildings by using mathematical objects known as "polytope complexes" or "linear complexes" which are a special case of so-called "cw complexes". In this special case, the cells are polytopes, and of course in the building context, they are
of dimension three. In the introduction section, it is explained how such linear complexes are stored in tables. However, it is unclear to me, whether the authors are referring to the state of the art in the literature on explicit topological modelling of buildings, or whether these tables are already claimed to be part of their own contribution to the science of building information modelling.

In any case, it seems to me that the authors are not aware of developments in that field which have already been going on for more than 15 years. Namely, what they describe as a basis for generating the possible topological tables has already been used by other authors in order to represent topological building information in relational databases. The idea in fact comes from mathematics, more precisely the domain of algebraic topology, where the boundary information between cells of neighbouring dimensions in cw complexes is stored explicitly in matrices. The mathematical literature has been studying these objects for more than a hundred years. I therefore strongly recommend to include a discussion and references to that literature, e.g. contained in textbooks on algebraic topology. This is the mathematical basis for using the relational model to capture topological information of spaces, published 12 years ago.

The authors also claim that the topology tables are a lossless representation of linear complexes. In general, however, this is not true. But in the context of building models, this might be the case. Anyway, this important aspect should also be discussed in a revision of this article.

The actual contribution of this research is an algorithm for constructing all relational tables from the given cell-face, face-edge and edge-vertex tables, including its complexity. This is of interest, because those other relationships can be then given further information to store in a database. They prove that the complexity is linear in the number of cells, if the counts of faces and edges per cell are constant. They argue that this is the case in the building sector.
However, they should underline that this assumption is important, as otherwise, the complexity is not linear.

For more literature on explicit topological modeling of building and city models, cf. e.g.

Breunig, M.; Bradley, P. E.; Jahn, M.; Kuper, P.; Mazroob, N.; Rösch, N.; Al-Doori, M.; Stefanakis, E.; Jadidi, M. (2020). Geospatial Data Management Research: Progress and Future Directions. ISPRS International Journal of Geo-Information, 9 (2), 95

T. Krämer; W. Huhnt. Topological Information in Geometrical Models of Buildings. International Workshop on Computing in Civil Engineering 2009

R. Aish. W. Jabi. S. Lannon. N.M. Wardhana. A. Chatzivasileiadi. Topologic Tools to explore Architectural Topology Advances in Architectural Geometry 2018, Chalmers University of Technology, Gothenburg, Sweden, 22-25 September 2018.

and the references contained in these articles.

I strongly recommend a major revision.

Author Response

The authors thank the reviewer for the evaluation of the submitted paper and for helpful comments and suggestions.

Following the written recommendations of the reviewers, as made available by the editor, the authors have modified the submitted paper and have made additions that have been suggested by the reviewers. In the attached of the modified paper, the modifications are marked in yellow color. The following remarks describe the considerations of the authors concerning the modifications and the additions.

1. The reviewer points out that it is unclear “whether the authors are referring to the state of the art in the literature on explicit topological modelling of buildings, or whether these tables are already claimed to be part of their own contribution to the science of building information modelling.”

The authors do not claim that topological tables are part of their contribution to the science of building information modelling. Such tables are part of the established state of the art, for example in the Industry Foundation Classes IFC. The authors wish to show that the application of the existing concept of topological tables was a basic step in the transition from static line drawings to dynamic digital models, to present an efficient algorithm for the implementation of the concept in digital models, and to give reasons for limits to the applicability of topological tables. In order to clarify this intention, the following text has been added to the paper below figure 2:

“The contact between the domains of the complex is described with the established concept of topological tables.”

2. The reviewer states: “In any case, it seems to me that the authors are not aware of developments in that field which have already been going on for more than 15 years. Namely, what they describe as a basis for generating the possible topological tables has already been used by other authors in order to represent topological building information in relational databases. The idea in fact comes from mathematics, more precisely the domain of algebraic topology, where the boundary information between cells of neighbouring dimensions in cw complexes is stored explicitly in matrices. The mathematical literature has been studying these objects for more than a hundred years. I therefore strongly recommend to include a discussion and references to that literature, e.g. contained in textbooks on algebraic topology. This is the mathematical basis for using the relational model to capture topological information of spaces, published12 years ago.”

The authors are aware of the efforts by other authors to represent topological building data in relational databases. Relational data bases are an alternative to the object-oriented method of data management used by the authors. The state of the art strongly favors the object-oriented methods, for example in the Industry Foundation Classes and their many implementations in the established commercial software for the planning, architectural and building sectors. A scientifically founded comparison of the two methods goes beyond the scope of this paper.

In order to draw the attention of the reader to the option of using relational data bases, the two references that the reviewer has kindly provided have been added to the list of references:

  1. Breunig, M., Bradley, P.E., Jahn, M. Geospatial Data Management Research: Progress and Future Directions. ISPRS Int. J. Geo-Inf. 2020, 9(2), 95. DOI https://doi.org/10.3390/ijgi9020095
  2. Aish, R.; Jabi, W.; Lannon, S. Topologic: tools to explore architectural topology. In Proceedings of the Advances in Architectural Geometry. Gothenburg, Sweden, 22-25 September 2018.

The two references are cited in the paper in the following sentence, which has been included in the paragraph following figure 9:

“The option of using relational data bases rather than the object-oriented data structures presented by the authors is presented in references [44] and [45].”

3. The reviewer states:  ”The authors also claim that the topology tables are a lossless representation of linear complexes. In general, however, this is not true. But in the context of building models, this might be the case. Anyway, this important aspect should also be discussed in a revision of this article”.

The authors have not been able to find a text in the paper that in their opinion gives rise to the stated concern of the reviewer. The authors do not claim that the topology tables are a lossless representation of linear complexes. On the contrary, section 4 of the paper explicitly treats examples of topological information of linear complexes that is not contained in topological tables. The authors have strengthened the presentation of their view on this topic by adding the paragraph following figure 9.

4. The authors appreciate the positive evaluation of their algorithm by the reviewer. As the reviewer has suggested, the authors have underlined that the complexity of the algorithm is only linear if the assumption is satisfied that the number of edges per face and the number of faces per cell are constant. The necessary text has been added in the abstract of the paper and following equation (2) in the text.

Round 2

Reviewer 2 Report

The objections raised in the previous version are now duly met. The article can now be accepted in my point of view. The reviewer thanks the authors for the helpful clarifications contained in their response.